

# Dinosaur origin of egg color: oviraptors laid blue-green eggs

Jasmina Wiemann[1,2], Tzu-Ruei Yang[1], Philipp N. Sander[3,4], Marion Schneider[5], Marianne Engeser[6], Stephanie Kath-Schorr[3], Christa E. Müller[5] and P. Martin Sander[1]

[1] Division of Palaeontology, Steinmann Institute of Geology, Mineralogy and Palaeontology, University of Bonn, Bonn, Germany
[2] Department of Geology & Geophysics, Yale University, New Haven, CT, United States of America
[3] Life and Medical Sciences Institute, University of Bonn, Bonn, Germany
[4] Department of Chemistry, University of California, Berkeley, United States of America
[5] Pharmaceutical Institute, Pharmaceutical Chemistry I, University of Bonn, Bonn, Germany
[6] Kekulé Institute for Organic Chemistry and Biochemistry, University of Bonn, Bonn, Germany

Corresponding author
Jasmina Wiemann,
jasmina.wiemann@yale.edu

## ABSTRACT

Protoporphyrin (PP) and biliverdin (BV) give rise to the enormous diversity in avian egg coloration. Egg color serves several ecological purposes, including post-mating signaling and camouflage. Egg camouflage represents a major character of open-nesting birds which accomplish protection of their unhatched offspring against visually oriented predators by cryptic egg coloration. Cryptic coloration evolved to match the predominant shades of color found in the nesting environment. Such a selection pressure for the evolution of colored or cryptic eggs should be present in all open nesting birds and relatives. Many birds are open-nesting, but protect their eggs by continuous brooding, and thus exhibit no or minimal eggshell pigmentation. Their closest extant relatives, crocodiles, protect their eggs by burial and have unpigmented eggs. This phylogenetic pattern led to the assumption that colored eggs evolved within crown birds. The mosaic evolution of supposedly avian traits in non-avian theropod dinosaurs, however, such as the supposed evolution of partially open nesting behavior in oviraptorids, argues against this long-established theory. Using a double-checking liquid chromatography ESI-Q-TOF mass spectrometry routine, we traced the origin of colored eggs to their non-avian dinosaur ancestors by providing the first record of the avian eggshell pigments protoporphyrin and biliverdin in the eggshells of Late Cretaceous oviraptorid dinosaurs. The eggshell parataxon *Macroolithus yaotunensis* can be assigned to the oviraptor *Heyuannia huangi* based on exceptionally preserved, late developmental stage embryo remains. The analyzed eggshells are from three Late Cretaceous fluvial deposits ranging from eastern to southernmost China. Reevaluation of these taphonomic settings, and a consideration of patterns in the porosity of completely preserved eggs support an at least partially open nesting behavior for oviraptorosaurs. Such a nest arrangement corresponds with our reconstruction of blue-green eggs for oviraptors. According to the sexual signaling hypothesis, the reconstructed blue-green eggs support the origin of previously hypothesized avian paternal care in oviraptorid dinosaurs. Preserved dinosaur egg color not only pushes the current limits of the vertebrate molecular and associated soft tissue fossil record, but also provides a perspective on the potential application of this unexplored paleontological resource.

## INTRODUCTION

Birds offer the most diverse displays of color and shape among modern vertebrates (*Stoddard & Prum, 2011*). In contrast to their closest modern relatives, the crocodiles, avian eggs range widely in size and shape, and the avian key innovation seems to be variation in color (*Cassey et al., 2012*). Nature's repertoire ranges from immaculate (homogeneously colored) reddish brown and white eggs in chicken (*Gallus domesticus*), to light beige with dark brown maculation (speckling) in the oystercatcher (*Haematopus ostralegus*), to light blue in the American robin (*Turdus migratorius*), to the intensive bluish-green of emu eggs (*Dromaius novaehollandiae*) at the end of the spectrum (*Cassey et al., 2012*).

All other amniotes, including non-avian reptiles and monotreme mammals, lack eggshell color (*Packard & Seymour, 1997*). Non-avian amniotes can protect their eggs by burying them (*Leighton, Horrocks & Kramer, 2009*), or by continuously guarding the nest (*Komdeur & Kats, 1999*). Colored eggs are present in most modern birds which build open nests, as the eggs are vulnerable due to periods without parental guarding (*Komdeur & Kats, 1999*; *Gillis et al., 2012*). Birds with biparental brooding behavior minimize the periods during which a clutch is unattended and vulnerable, and sometimes reduce egg coloration (*Komdeur & Kats, 1999*). Complete reduction of eggshell coloration is observed in many cavity nesting and cave-breeding birds (*Hewitson, 1864*) confirming the visual signaling component of egg color function (*Castilla et al., 2007*). Visual signaling resulting in camouflage of an egg clutch is largely dependent on eggshell coloration relative to the color shade of the nesting background (*Wallace, 1890*; *Stoddard et al., 2016*). Such negative signaling also offers protection against brood parasitism because of, e.g., elaborate egg color innovations that allow recognition of parasite eggs among the clutch (*Newton, 1896*). In addition to signaling, numerous other functions of egg color pigments have been proposed, such as antimicrobial effects (*Ishikawa et al., 2010*), protection from solar radiation (*Lahti, 2008*), and eggshell mechanical reinforcement (*Gosler, Higham & Reynolds, 2005*). In a phylogenetic analysis of egg shell coloration and color patterning, *Kilner (2006)* concluded that the ancestral egg color of avian of extant birds was white, and that egg coloration evolved multiple times within crown birds (see *Kilner, 2006*).

This phylogenetic inference ignores the fact that many birds with white eggs, including ostrich (*Struthio*), rhea (*Rhea*) and elephant bird (*Aepyornis*) contain minor amounts of eggshell pigment, and that their reduced pigment most likely represents an evolutionary reaction to the brooding-based reduced selection pressure on coloring their eggs (*Kennedy & Vevers, 1975*).

We hypothesize egg coloration evolved after the switch from burying eggs to building an open and exposed nest (consistent with *Gillis et al., 2012*). Selection for egg color would only have occurred after the eggs themselves became visible to parents, conspecifics, predators, or parasites (*Kilner, 2006*).
While most dinosaurs buried their eggs (reviewed by *Varricchio & Jackson (2016)*), there is ample evidence that bird-like, non-avian eumaniraptoran dinosaurs, i.e., Oviraptorosauria, Dromaeosauridae, and Troodontidae, built open ground nests with at least partially exposed eggs (*Varricchio & Jackson, 2016*; *Norell et al., 1995*). Partial exposure accounts for the arrangement of their strongly elongated eggs stacked and partially buried almost vertically in the nest material in circular layers either with (seemingly primitive) or without (derived) an empty space in the center of the nest (*Norell et al., 1995*). Late Cretaceous oviraptorosaurid eggs from China and Mongolia ((*Norell et al., 1995*); reviewed in (*Varricchio & Jackson, 2016*)) are frequently preserved, and because they are laid in exposed, partially open nests (*Norell et al., 1995*), they may have been pigmented like many bird eggs.

Only two chemical compounds act as avian eggshell pigments and give rise to virtually all observed bird egg colors and patterns–the tetrapyrrols protoporphyrin (PP) and biliverdin (BV) (*Kennedy & Vevers, 1975*). Both are participants in the vertebrate heme cycle: protoporphyrin is a reddish-brown heme precursor, while biliverdin is a blue–greenish heme catabolite (*Ryter & Tyrrell, 2000*). In contrast to the linear BV, the cyclic PP exhibits a stronger resonance stabilization (*Falk, 1964*). Due to their different metabolic functions, PP and BV have distinctively different chemical properties: while PP is lipophilic and rather unreactive, BV is much more reactive due to its hydrophilic, oxidated, linear arrangement (*Gorchein, Lim & Cassey, 2009*). BV is distributed throughout the entire thickness of the prismatic layer in eggshells, most probably linked to matrix proteins, while PP is currently thought to be present in high concentrations only in the outermost eggshell cuticle layer (*Wang et al., 2009*). Related avian pigments are incorporated into integumentary structures, such as, for example, the PP relative uroporphyrin III which occurs as red turacine pigment in the feathers of herbivorous, musophagid birds (*Rimington, 1939*). Structurally, the two avian eggshell pigments show a similarity to the most common vertebrate pigments eu- and pheomelanin, in being N-heterocycles (*Watt, Bothma & Meredith, 2009*) although their origins are very different in vertebrate secondary metabolism (*Kubo & Furusawa, 1991*).

In terms of pigment taphonomy, the oldest confirmed record of eggshell PP and BV traces is in subfossil moa eggs from New Zealand (*Igic et al., 2009*) which demonstrates their preservation potential on a time scale of $10^3$ years, but also the loss of the more labile BV through time due to degradation processes, dissolution and transport via percolating aqueous fluids. Other related biomolecules with a reported fossil record are hemes (*Greenwalt et al., 2013*), and chlorophylls (*Leavitt, 1993*). Both have been identified with minimal or no diagenetic alteration from Mesozoic and Cenozoic fossil deposits, supporting the possibility of eggshell pigment preservation in fluvial or alluvial oxidative deposits such as those from eastern and southern China.

We identified unmodified, preserved PP and BV eggshell pigments in all three oviraptorid samples and proved that these eggshells were the sole source of pigments by demonstrating the absence of BV and PP in the investigated sample of adjacent sediment. We also provide support for the preservation of dinosaur eggshell cuticle (previously suggested by *Mikhailov, Bray & Hirsch, 1996*; *Schweitzer et al., 2002*; *Varricchio & Jackson, 2004*) based on cuticular PP storage. These observations phylogenetically extend the presence of an

avian-like eggshell pigmentation back to oviraptorids. Using the known concentrations of our commercial pigment standards and PP and BV concentrations in emu eggshell as a sensitivity control and quality marker for our analytic system, we reconstructed a visually evident blue–green egg color for *Heyuannia* incorporating experimental and taphonomic corrections. Our eggshell zonal porosity reevaluation reconstructed an at least partially open nest for the oviraptorid *Heyuannia huangi* (based on *Varricchio et al., 2008*; *Deeming, 2006*; see Supplemental Information).

## MATERIALS AND METHODS

### Extant eggshell material
The emu eggshells (*Dromaius novaehollandiae*) were produced by captive birds and are stored in the ZFMK collections (ZFMK uncat.). Emu eggshell is reported to contain minimal amounts of PP in addition to some of the highest naturally occurring amounts of BV (*Gorchein, Lim & Cassey, 2009*).

### Fossil eggshell material
We sampled three oviraptorid *Macroolithus yaotunensis* eggs from the collections of the NMNS and STIPB covering three geographically and taphonomically distinct Chinese deposits. Investigated specimens were collected in the Liguanqiao Basin near Nanyang in the province of Henan (STIPB E54/1), from the Hongcheng Basin in the province of Jiangxi (NMNS CYN-2004-DINO-05/I), and the Nanxiong Basin in province of Guangdong (STIPB E54/3). Detailed descriptions of the localities in context of geological and taphonomic settings are included in the Supplemental Information. Macroscopically, all oviraptorid eggshell samples had a blackish to blackish-brownish (after cleaning them from adherent sediment) color, revealing a very subtle shimmer of blue–green at angled light conditions.

Historically, samples derived from the Liguanqiao Basin are Late Cretaceous in age and derived from the fluvial/alluvial deposits (red sandstones) of the Hugang Formation. They have been housed in STIPB since 1983 and were previously described by *Erben (1995)*. Preserved oviraptorid eggs from the Hongcheng Basin in the province of Jiangxi were obtained from the Late Cretaceous Tangbian Formation which comprises fluvial red sandstones. The Hongcheng Basin and the Nanxiong Basin may belong to the same extended basin complex (*Liu, 1999*). The Late Cretaceous strata of the Nanxiong Basin are divided into the Yuanpu Formation and the overlying Pingling Formation. The Yuanpu Formation, which might be correlated with the fossil-rich Mongolian Djadochta Formation, yielded our eggshell samples from alluvial sediments (red silt-sandstones), and is dated as Maastrichtian in age (*Zhao et al., 1991*).

One of two preserved complete eggs (Fig. 1A) from the Chinese province of Jiangxi (NMNS CYN-2004-DINO-05) which were previously assigned to the oviraptorid egg parataxon *Macroolithus yaotunensis* was sampled over four zones of the egg (Figs. 1B and 2B), prepared for histology, and then used for porosity measurements (see Supplemental Information). These four zones represent the blunt, middle, and acute parts of the egg, and were separated to approach zonal differences in porosity values which were tested for

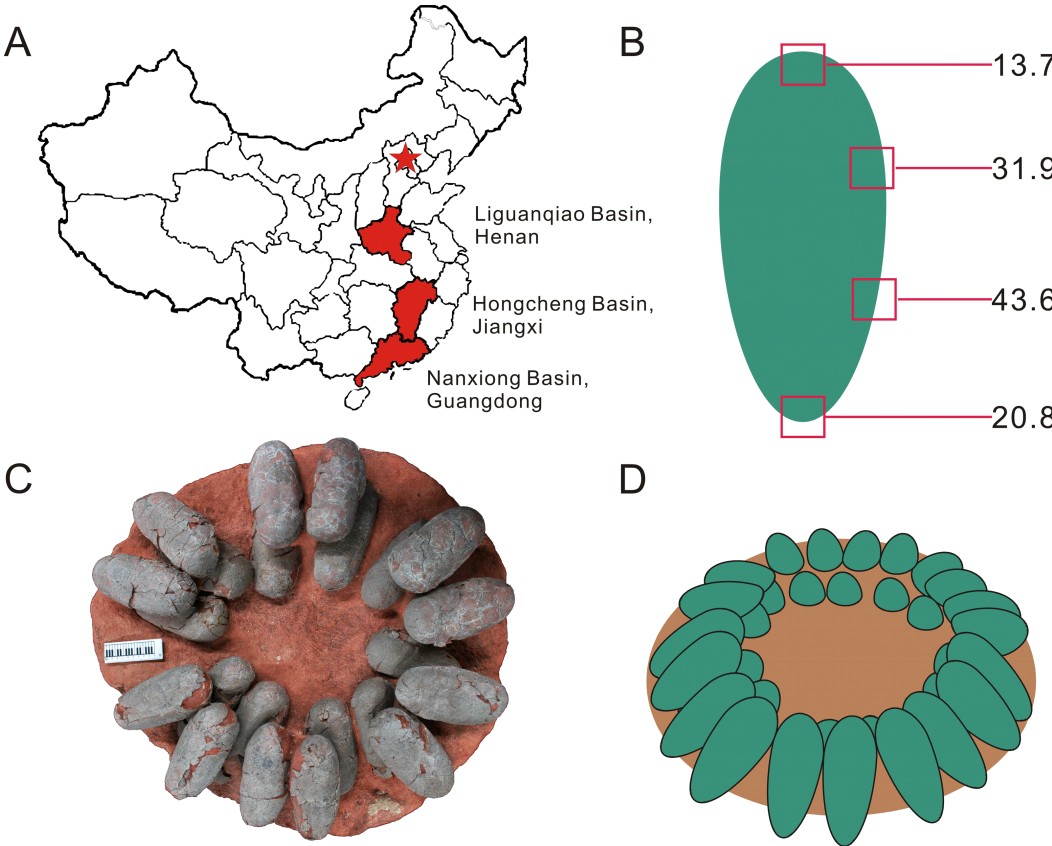

**Figure 1** **Provenance of *Heyuannia* eggshell, reconstructed zonal egg water vapor conductance, oviraptor clutch structure, and corrected, reconstructed egg color.** (A) Geographical map of China. The capital city, Beijing, is indicated by the red star. Red shaded provinces indicate the three different localities where the specimens were collected: the Liguanqiao Basin in Henan, the Hongcheng Basin in Jiangxi, and the Nanxiong Basin in Guangdong (see Supplemental Information). (B) The reconstructed color and average zonal water vapor conductance of the left Jiangxi *Heyuannia huangi* egg (NMNS CYN-2004-DINO-05/I) calculated from BV and PP concentrations and porosity measurements (see Supplemental Information). (C) Top view of an oviraptor clutch (PFMM 0010403018). This clutch illustrates how eggs are arranged in pairs with their blunt ends pointing to the clutch center. The eggs are arranged in layers separated by sediment. (D) Reconstruction of a partially open oviraptorid nest. Note that the original inclination of the eggs would have been steeper than their preserved attitude (C) due to sediment compaction.

maximum porosity at the mid portions to indicate egg storage in an open nest (based on *Varricchio et al., 2008*). Measured porosity values were compared to published dinosaur and avian porosity patterns and used to calculate the eggshell water vapor conductivity. Samples for chemical analyses were taken separately. Sediment adhering to the complete eggs (NMNS CYN-2004-DINO-05/I) was sampled additionally to confirm that we are not dealing with wholesale sample contamination. A single sediment sample (red silty sandstone) was available to test against wholesale contamination with BV and PP of the sample since only the complete oviraptorid eggs from the province of Jiangxi provided original, attached matrix sediment. The two-remaining fossil oviraptor eggshell samples represent isolated fragments freed of original matrix.

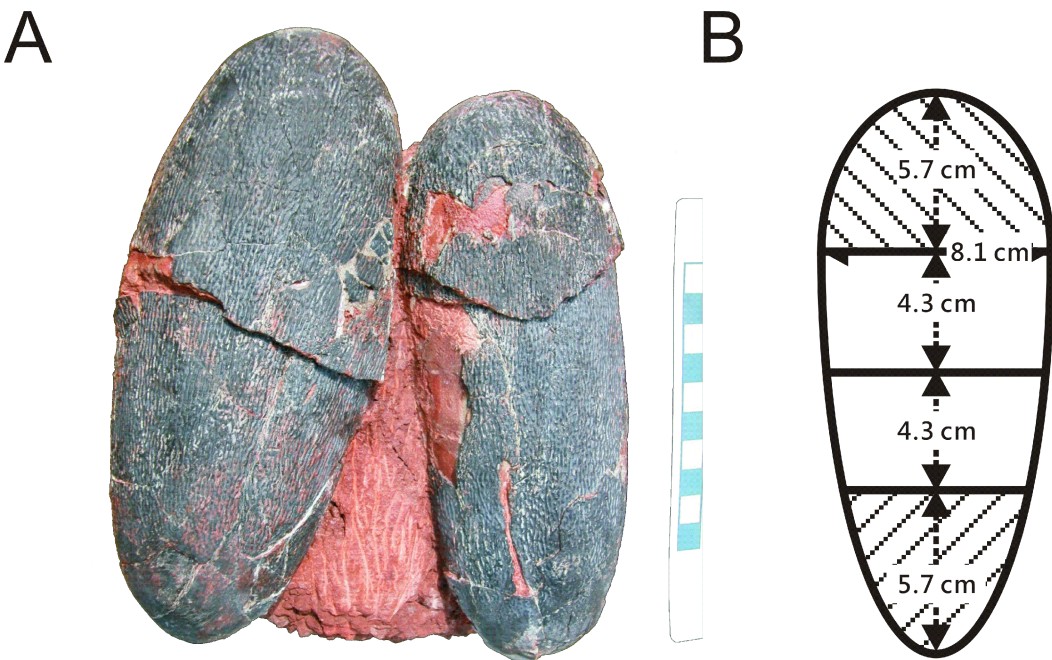

**Figure 2** (A) Pair of oviraptorid *Heyuannia* eggs (NMNS CYN-2004-DINO-05) from the Chinese province of Jiangxi before sampling. Porosity measurements and calculations of water vapor conductance are based on these eggs. Pieces of eggshell from each of the four zones depicted in (B) were used in porosity measurements. (B) Egg model separated into four zones used for zonal porosity measurements. Therefore, double half-prolate spheroids and cone models of the idealized egg were used to estimate the zonal surface areas to eventually approximate water vapor conductance. Zone 1 represents the blunt end of the egg, zones 2 and 3 the mid portions, and zone 4 represents the pointed end of the egg.

## METHODS

We used two commercial standards (biliverdin dihydrochloride and protoporphyrin IX, purchased from Sigma Aldrich), one extant bird eggshell sample (emu), the three fossil *Heyuannia huangi* eggshell samples, and one sediment sample (reddish sandstone) for High Performance Liquid Chromatography coupled to Electrospray Ionization Quadrupole Time-of-Flight Mass Spectrometry (HPLC ESI Q-ToF MS).

The basis for this sample selection was (1) to demonstrate reproducibility in three fossil eggshell samples, (2) to exclude the possibility of wholesale contamination due to sample or system exogenous PP or BV input by analyzing sediment adhering to a sample of eggshell, (3) to provide a sensitivity control and quality marker of the analytical routine based on precise detection and quantification of emu eggshell (ranging from the upper (BV) to the lower (PP) detection limit in terms of pigment concentrations), and (4) to generate a calibrated concentration signal for quantification based on the known concentrations of the commercial standards.

Adhering sediment and other superficial contaminations were chemically removed from all eggshell samples by a boosted decalcification of the outermost sample surfaces. 500 μL of disodium EDTA solution (100 mg/mL), adjusted to a pH of 7.2, were added to the 180–562 mg eggshell samples and sediment control sample, each of which was stored in

1 mL Eppendorf tube. Samples were incubated for 5 min and then transferred to fresh Eppendorf tubes. The decalcification residue was discarded.

Pigment decalcification was performed immediately after preliminary cleaning of the samples. The three oviraptorid eggshell samples, the emu eggshell, and the sediment control sample were incubated again in the EDTA solution which was already used for initial cleaning, this time for 5 min. During this 5 min, the sample tubes were vortexed three times for 1 min. Vortexing was performed with uncapped tubes to allow outgassing of the carbon dioxide generated. After 5 min of incubation in the EDTA decalcification solution, all sample tubes were centrifuged at 15,000 g for 1 min. After centrifuging, the supernatant solutions were collected in separate tubes, while the decalcified sample precipitates were filled up with fresh EDTA solution from the stock. Incubation of 5 min followed, including vortexing $3\times$ in uncapped tubes, as in the previous step. The samples were centrifuged again at 15,000 g for 1 min, supernatants were collected, and the precipitates filled up with fresh EDTA solution. As in the previous step, incubation of 5 min including vortexing of the uncapped sample tubes followed. After a final round of centrifuging for 1 min, supernatant solutions were collected, and the partially decalcified precipitates were used for the final pigment extraction. 1 mL of acetonitrile/acetic acid (4:1, *v/v*) was added to the decalcified sample pellets for 10 min of incubation, including 2 min of vortex-mixing. Afterwards the sample tubes were centrifuged at 15,000 g for 2 min, and the supernatant solution holding the pigment extract were transferred into fresh Eppendorf tubes, and stored in a dark environment at 4 °C. The commercial standards were dissolved in the same acetonitrile-acetic acid solution (4:1, *v/v*) and stored with the sample pigment extracts.

The filtered extracts and commercial standard solutions were stored less than 24 h before they were injected into an HPLC Dionex Ultimate 3000 (Thermo Scientific) separating sample compounds by using a EC50/2 Nucleodur C18 Gravity 3 $\mu$m column (Macherey-Nagel). Reverse-phase HPLC was run at a flow rate of 0.3 mL/min. HPLC was started at 90% $H_2O$ containing 0.1% acetic acid. The gradient started after 1 min and reached 100% acetonitrile after 14 min. For an additional 7 min, the column was flushed with 100% acetonitrile (containing 0.1% acetic acid). For the biliverdin analysis, 15 $\mu$L sample solution was injected, and 20 $\mu$L for the protoporphyrin detection. 2 min of washing runs between each sample extract cleaned the entire system. The liquid chromatography system was coupled to a micrOTOF-Q mass spectrometer (Bruker) with an electrospray ionization (ESI) source inducing positive ionization.

Data were collected in positive full scan MS mode over the range of 50–1,000 m/z, using a capillary voltage of 4.5 kV and an end plate offset of $-500$ V. The dry heater of the ESI source was set at 200 °C. Nitrogen desolution and nebulizer gas flow was 10.0 L/min; the nebulizer was run at 2.2 bar. Time-of-Flight (TOF) detection allowed the determination of the accurate masses of biliverdin and protoporphyrin.

## RESULTS

We reliably identified both BV, as $[M + H]^+$ with 583.2520 m/z (calculated mass: 583.2551 g/mol) after 8 min retention time, and PP, as $[M + H]^+$ complex with

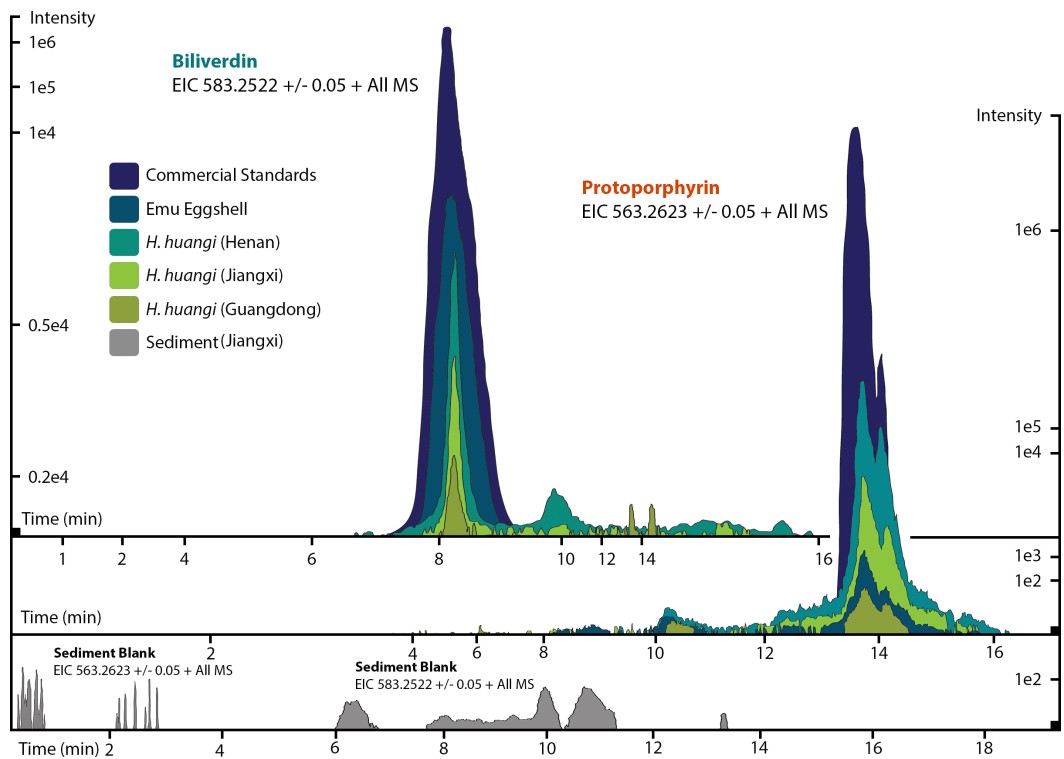

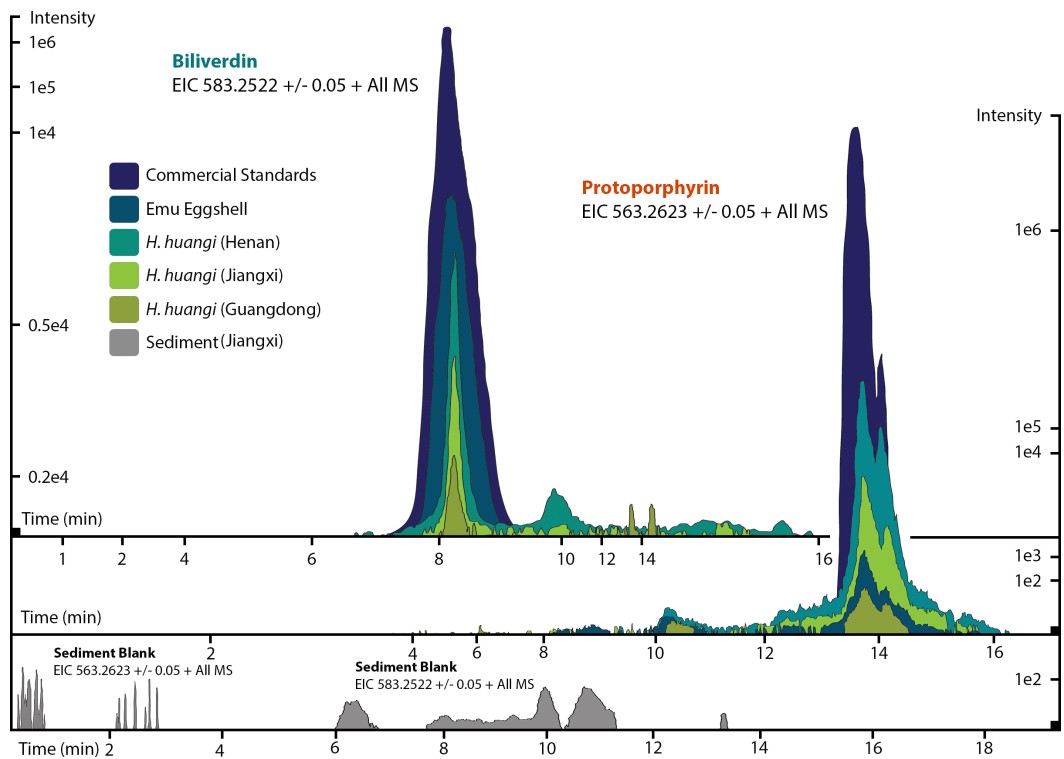

**Figure 3 ESI (+) MS extracted ion chromatograms (EICs) for mass 583.2520 ± 0.01/0.05 m/z, indicative of BV, and mass 563.2653 ± 0.01/0.05 m/z, indicative of PP.** We identified BV and PP by retention time, exact mass and isoform/tautomer separation. EICs for 583.2520 ± 0.01/0.05 m/z are depicted for the commercial BV standard, emu eggshell, extracts of *Heyuannia huangi* eggshells derived from the Chinese provinces Henan, Jiangxi, and Guangdong and the sediment control extract from Jiangxi. Unmodified BV elutes after 8 min retention time, and was proven present for the biliverdin commercial standard, the emu eggshell, and the three oviraptorid eggshells. The sediment sample was used as control for contamination, and its EIC does not show a peak after 8 min retention time, proving the absence of biliverdin in the sediment sample and the originality of biliverdin detected for the eggshell samples. EICs for 563.2653 ± 0.01/0.05 m/z are depicted for the commercial PP standard, emu eggshell, *Heyuannia huangi* eggshell derived from the Chinese provinces Henan, Jiangxi, and Guangdong, and the sediment sample from Jiangxi. Unmodified PP elutes after 14 min and was proven present for the PP commercial standard, the emu eggshell, and the fossil oviraptorid eggshells. There is no peak in the PP EIC for the sediment sample after 14 min, what proves the absence of PP in the sediment control sample, and the originality of PP in the eggshell samples. Peak intensity correlates with pigment concentrations in the extracts.

563.2623 m/z (calculated mass: 563.2653 g/mol) after 14 min retention time (Fig. 3A) in the commercial standard solutions, the emu eggshell, and the three fossil oviraptorid eggshell samples. The commercial standard solutions of known concentrations were used to identify the chemotrait-specific, diagnostic retention times for BV and PP on the chromatographic column. Elution of BV was consistent after 8 min, while elution of PP delayed consistently until 14 min of the run mobile phase gradient (consistent with *Igic et al., 2009*). A second compound-diagnostic trait was provided by ionization after elution from the chromatographic column, followed by exact mass determination of the [M + H]$^+$ ion complexes. Differing behavior of BV and PP native to eggshells compared to commercially purified BV and PP was ruled out by identical retention times, number of

isoforms/tautomers, and exact mass peaks of emu eggshell BV and PP and the commercial standard solutions. Furthermore, maximum sensitivity of the HPLC ESI MS system was demonstrated by precise detection of pigment concentrations in the emu eggshell solution which approached the upper (BV) and the lower (PP) detection limits. The extracted ion chromatograms (EICs) for BV and PP obtained from the sediment sample whole ion mass spectrum yielded signals within broader tolerances of the BV and PP exact masses, but no peaks corresponding to the commercial standard calibrated retention times were identified. Absence of a diagnostic mass peak after 8 min retention time on the column for BV, and after 14 min retention time on the column for PP, is a significant demonstration of absence of trace amounts BV and PP in the sediment sample. Thereby, contamination of the samples or the detection system was excluded and originality of the detected pigments in the oviraptorid eggshell samples is guaranteed.

Quantification of the detected pigment concentrations based on commercial standard calibration was determined by application of an experiment-empirical correction for the extraction loss of BV due to its increased hydrophily of the fossil oviraptor eggs.

We found the highest preserved concentrations of BV in the eggshells from Henan (6 nmol/g), followed by the eggshells from Jiangxi (2 nmol/g) and those from Guangdong (1 nmol/g). The preserved PP concentrations ranged from 2 nmol/g in both the Henan and Jiangxi eggshells to 1 nmol/g in the Guangdong eggshells. Our empirical correction applied to the fossil eggshell samples yielded very realistic pigment concentration estimates for the emu eggshell of 2 nmol/g PP, and 266 nmol/g BV which fall into the reported range of emu eggshell pigment concentrations in the scientific literature (Table S1). The bluish shimmer of the fossil oviraptorid eggshells suggests generally higher BV concentrations than those we detected, as also found in a similar study on pigment preservation in subfossil moa eggshells using the same methodology. The color of the fossil eggs suggested higher BV concentrations than the authors managed to detect. Taken together, our study and the study by *Igic et al. (2009)* imply that bluish or greenish coloring degradation products of BV remain which therefore shows slightly different chemical properties and different exact masses, and is thus not detected by an LC MS system targeting unmodified compounds measurable against commercial standards.

To demonstrate the perceivability of a visual color signal based on the detected pigment concentrations in the oviraptorid eggshells, we plotted our pigment concentrations into the comprehensive pigment concentration-color matrix of *Cassey et al. (2012)*. Our three dinosaur egg color data points fall in the visibly olive-green color range between *Haliaetus albicilla* and *Circus aeruginosus* (*Cassey et al., 2012*). They plot in the cluster of unspotted eggs, suggesting an immaculate, homogenous coloration. No patterns were visible in the fossil eggs (Fig. 1).

Since the preserved fossil oviraptor eggshell color suggests originally higher BV concentrations, taphonomy needs to be considered to generate a realistic, native oviraptor egg color reconstruction. Because BV is more reactive and more hydrophilic, and thus soluble in sediment-percolating aqueous fluids, the concentrations of unmodified, preserved pigments after at least 66 million years of sedimentary burial are much more likely to be significantly lowered than those of the more stable, hydrophobic PP (*Falk,*

*1964*). Therefore, the taphonomic projections of our preserved pigment concentrations in the avian egg color space (*Cassey et al., 2012*) realistically lift the investigated oviraptor egg colors significantly towards much higher BV values, while the shift towards increased PP values would be only minimal. However, our fossil oviraptor eggs would remain deeply nested within the area of unspotted eggs (based on *Cassey et al., 2012*). Such an additional taphonomic correction of the reconstructed egg color approximates an intensively blue–greenish oviraptorid egg color. Whether the differences in preserved pigment concentrations between the three fossil oviraptor egg samples from different localities reflect intraspecific variation in egg color or different taphonomic conditions in the deposits cannot be reliably assessed at this point and requires future investigations. Since these differences in preserved pigment concentrations in the oviraptorid eggshells affect the BV values much stronger (range 6–1 nmol/g) than they affect the PP values (range 2–1 nmol/g), we assume that differences in color are more likely to be taphonomic. *In vivo* intraspecific variation of egg color would most likely affect BV and PP concentrations equally, while taphonomic effects affect BV concentrations much stronger than PP concentrations (*Falk, 1964*). Also, the original egg color is overprinted by a generally blackish-brownish hue (Fig. 1). This brownish discoloration traces back to preserved, oxidatively crosslinked eggshell organic matrix proteins of the AGE/ALE-type (*Wiemann et al., 2016*).

Our reconstruction of colored eggs for oviraptors is consistent with our reevaluation of the oviraptorid nesting mode: we consider oviraptor eggs as lying at least partially open in the nest (consistent with (*Norell et al., 1995*)). This reconstruction is based on the estimated water vapor conductance of 108.66 mg $H_2O$ day$^{-1}$ Torr$^{-1}$ for the *Heyuannia* egg NMNS CYN-2004-DINO-05/I from the province of Jiangxi (Fig. 1A). This value is calculated from the four zonal conductances deduced from zonal porosity counts (Supplemental Information). The highest values for shell porosity were found in the middle portion of the oviraptor egg (zones 2 and 3), and especially pronounced in zone 3 with a resultant conductance of 43.6 mg $H_2O$ day$^{-1}$ Torr$^{-1}$ (Table S5). The pointed end (zone 4) which is stuck in the nest material, has a calculated conductance of 22.88 mg $H_2O$ day$^{-1}$ Torr$^{-1}$, while the exposed blunt end (zone 1) has a calculated conductance of 13.77 mg $H_2O$ day$^{-1}$ Torr$^{-1}$ (Table S5).

## DISCUSSION

Some aerobic microorganisms are known to produce protoporphyrin as an intermediate of their cytochrome c biosynthesis, and they also use biliverdin which represents a highly-conserved metabolite. Thus, a microbial origin of the pigments that we detected from our oviraptorid egg samples needs to be excluded. We use the double-checking LC MS approach (based on *Gorchein, Lim & Cassey, 2009*) applied to sediment adherent to the oviraptorid eggshells from Jiangxi to reliably exclude any wholesale sample contamination, contamination of the LC MS system, of used laboratory equipment or chemicals. Absence of any traces of BV or PP in our sediment control sample proves the cleanliness of our analytical system and supplies, as it proves absence of whole sample contamination based on

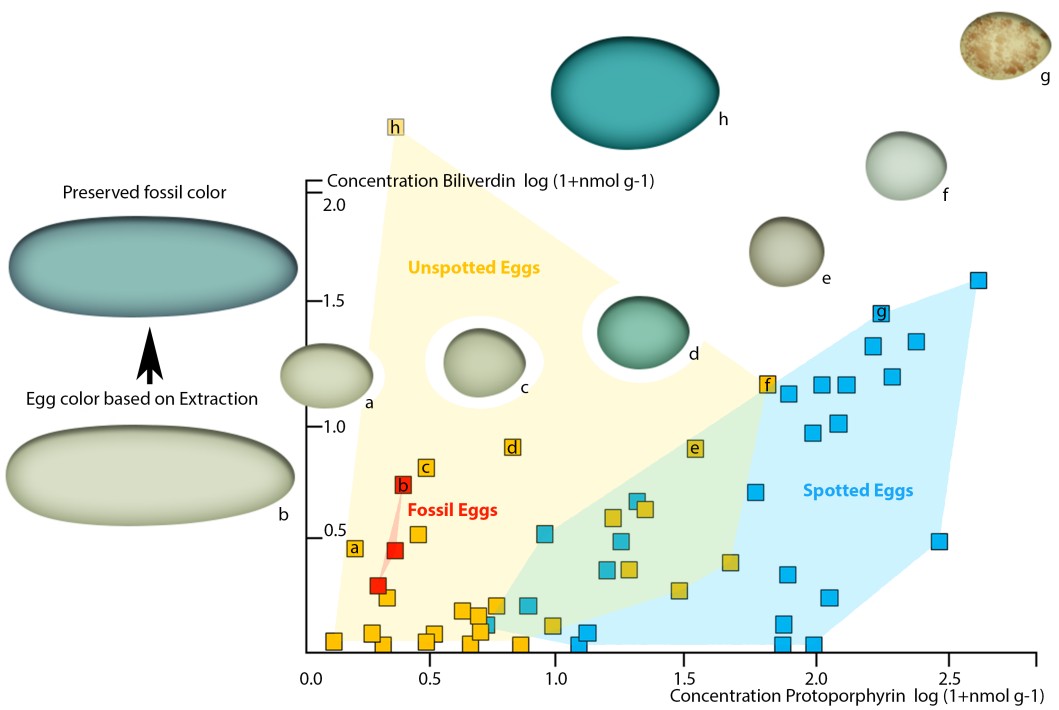

**Figure 4** **Plot of oviraptor egg and different avian egg biliverdin concentrations against their proto-porphyrin concentrations with example for the overall color impression based on the dataset published by** *Cassey et al. (2012)*. Avian eggs cluster together moderately separated into unspotted (yellow cluster, yellow squares) and spotted (blue cluster, blue squares) eggs. Fossil oviraptorid eggs span a color space (red cluster) and are represented by red squares, all nested within definitely unspotted eggs. The color examples extracted from *Cassey et al. (2012)* indicated for the preserved oviraptor egg pigments concentrations a visibly olive-green color (framed by (A) *Haliaetus albicilla*; (C) *Circus aeruginosus*). This color estimate lies still below the preserved bluish shimmer of the eggshells. Egg colors at cluster edges are extracted from *Cassey et al. (2012)* and labelled from (A)–(H).

potentially abundant microbes in the deposits. The only alternative source of the detected oviraptorid eggshell pigments would be eggshell-only contamination with microbially derived BV and PP (*Woodard & Dailey, 1995*). The preserved bluish-greenish shimmer of the investigated eggshells strongly argues against this. Also, the preserved pigment concentrations and differing BV and PP ratios eliminate the possibility of eggshell-only microbial contamination. The preserved concentrations fall within the empirical correction for pigment loss during the extraction routine (see Fig. 4). Thus, the most parsimonious conclusion is that we have documented oviraptorid eggshell pigments.

Our results push the origin of pigmented eggshells phylogenetically back to oviraptorid dinosaurs (see Fig. 5) (*Kilner, 2006*). To test for convergent evolution or homology of egg color between oviraptorids and crown birds, future investigations of pigmentation in other eumaniraptoran eggshells are required. Oviraptorid dinosaurs incorporated the same pigments and isoforms into their eggshells out of a pool of dozens of possible staining secondary metabolites (*McGraw, 2006*; *Hubbard et al., 2010*; *Stoddard & Prum, 2011*). If dinosaur egg color was a convergent character relative to the colored eggs of crown birds, the same color effect could have been achieved by use of different metabolites. Non-avian

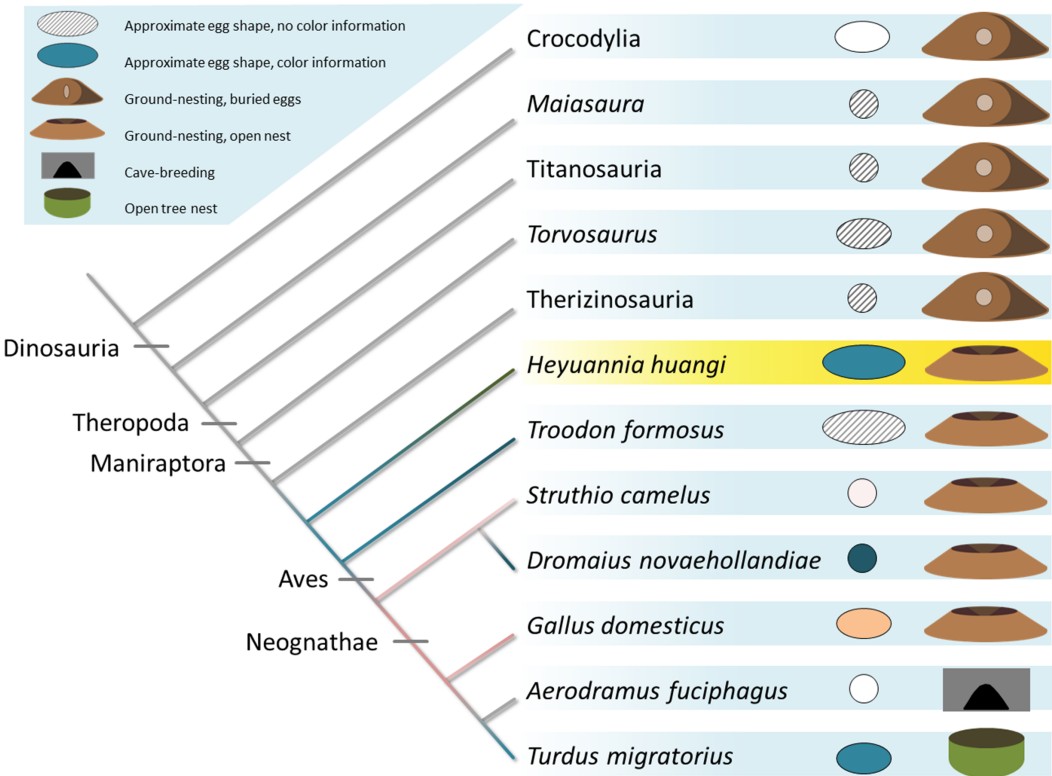

**Figure 5 Evolution of egg coloration (egg item), egg shape (egg item), and nesting type (nest item) in archosaurs.** Listed (as successive outgroups) are ornithischian dinosaurs, such as *Maiasaura*, for sauropods, such as titanosaurs, and theropod dinosaurs, including *Torvosaurus*, therizinosaurs, oviraptors, troodontids, and crown birds (*Varricchio & Jackson, 2016*). The topology of the tree is based on *Nesbitt, 2011* (Archosauria), *Sereno, 1999* (Dinosauria), *Carrano, Benson & Sampson, 2012* (Tetanurae), *Turner, Makovicky & Norell, 2012* (Paraves), and *Prum et al., 2015* (Aves). Porosity data and clutch structure indicate a fully buried nesting type for most dinosaur taxa (*Varricchio & Jackson, 2016*). Buried eggs which are indicated by the nesting item next to the egg item for each terminal taxon appear to be plesiomorphic. Oviraptorid dinosaurs, here represented by *Heyuannia huangi*, built at least partially open nests, concurrent with the phylogenetically most basal appearance of PP and BV in dinosaur eggshell (egg item). In modern birds, eggshell pigmentation varies with nesting microenvironment. Presence and kind of the (more abundant) eggshell pigment is represented by the color of the cladogram branch, and the egg item next to each terminal taxon. Beige egg colors indicate PP being more abundant, whereas bluish egg colors indicate BV being more abundant.

dinosaurs and modern birds apparently use the same molecules to create eggshell coloration. There are no studies available testing for pigment constraints during the shell formation based on simultaneous shell crystallite growth and pigment incorporation into the eggshell organic matrix. Also, potential constraints of metabolite transport mechanisms or *de novo* synthesis sites in the archosaur shell gland are yet unknown, as is metabolite permeability through the archosaur shell gland wall. In any case, we can infer based on our results that oviraptor shell glands generally worked in a similar way avian shell glands do: due to its hydrophilic behavior biliverdin is preferentially incorporated into the organic matrix of the eggshell prismatic zone which holds the calcite crystallites (*Falk, 1964*). Biomaterial studies suggested thioether linkage of BV to cysteine-rich proteins, allowing an energy-efficient

storage close to the ion lattice of calcite crystals (*Lamparter et al., 2004*; *Salewski et al., 2013*). Based on this presumably general storage mechanism of BV in biocomposite materials, we infer a similar storage in oviraptor eggshells. PP is secreted during the final steps of eggshell formation, while the storage mechanism is yet unclear. Due to its hydrophobic properties, PP is currently thought to occur in highest concentrations in the waxy eggshell cuticle (*Baird, Solomon & Tedstone, 1975*). Thus, evidence for PP in fossil oviraptorid eggshells strengthens the support for hypothesized preservation of eggshell cuticle through deep time (*Mikhailov, Bray & Hirsch, 1996*; *Schweitzer et al., 2002*; *Varricchio & Jackson, 2004*). Both pigments are supposed to be synthesized *de novo* in the shell gland tissue (*Wang et al., 2009*).

Combining our pigment analyses with the outcomes of studies on oviraptorid clutch arrangements (*Huh et al., 2014*; *Pu et al., 2017*), it can be summed up that oviraptorid eggs were most probably laid in overlapping circles, partially stuck in the nesting material, with their blunt ends exposed, pointing upwards in an almost vertical orientation. Partially exposed, blue–green eggs, stacked in circular layers, allow the inference of a similarly colored nesting material. In modern birds, blue–green eggs stored in ground nests are found in Casuariformes, including emus and cassowaries (*Coombs, 1989*). Emus and cassowaries lay their eggs in soil mounds covered with leaf litter and vegetation, so that egg color and nesting background match in tone (*Coombs, 1989*). Considering arid to subtropical paleoclimate reconstructions for Late Cretaceous China (*Hsu, 1983*), and the fluvial/alluvial red silt-sandstone sedimentology of the fossil laegerstaetten from Henan, Jiangxi and Guangdong provinces, vegetation coverage of the nest base comparable to emu and cassowary nests, adjacent to permanent or rather ephemeral river systems would offer an environment requiring blue–greenish egg colors for nest crypsis. An alternative explanation to only cryptic functions of blue–green eggs has been described for tinamous (*Tinamus major*): blue–green eggs in tinamous (*Tinamus major*) have been demonstrated to be non-cryptic and to be laid in environments where the egg color significantly contrasts the nesting background color (*Brennan, 2010*). In tinamous, nest predation depends not on egg color cues, but mainly on either visual or chemical parental cues during clutch incubation, and egg color is supposed to have evolved for intraspecific signaling (*Brennan, 2010*). Intraspecifically, blue–green egg color in modern birds has been associated with paternal care, communal nesting, and postmating sexual signaling (*Moreno & Osorno, 2003*; *Handford & Mares, 1985*). Paternal care was previously hypothesized to have dinosaur origin, tracing back to oviraptorid dinosaurs, and thus, poses a good fit our finding of oviraptorid blue–green egg color (*Varricchio et al., 2008*, but see *Birchard, Ruta & Deeming, 2013*). Communal nesting, as seen in polyandrous birds like emus and cassowaries, represents a reproductive strategy that might apply for non-avian dinosaurs, but has never been tested for (*Handford & Mares, 1985*). Preliminary studies based on eggshell chemical signatures identifying multiple maternal individuals contributing eggs to one clutch hint at communal nesting in oviraptorid dinosaurs (Yang et al., 2015, unpublished data). Finally, postmating sexual signaling according to the Sexual Signaling Hypothesis appears controversial in the ornithological literature (*Moreno & Osorno, 2003*; *Krist & Grim, 2007*). This hypothesis suggests intensive blue–green egg

color as a quality marker for maternal health and immunocapability (*Moreno & Osorno, 2003*). Post-mating sexual signaling then occurs due to BV being incorporated in high concentrations into the eggshell because it is dispensible to the maternal metabolism (*Moreno & Osorno, 2003*). The general idea goes back to the handicap hypothesis (*Roberts, Buchanan & Evans, 2004*), and assumes that the male is motivated by the confirmation of the female health status and thus, the offspring quality (*Moreno & Osorno, 2003*).

The similarity in reproductive strategies between crown birds and derived non-avian maniraptorans may reflect shared selective pressures: the recent discovery of the anseriform *Vegavis* in Late Cretaceous deposits from Antarctica provides evidence that crown birds coexisted with non-avian dinosaurs (*Clarke et al., 2016*). The presence of neognath birds in the Late Cretaceous implies that paleognath birds already diverged at this point (based on avian phylogeny by *Prum et al., 2015*). We would like to suggest new avenues of research based on the perspective of the potential coexistence, shared selection pressures, and niche competition of secondarily flightless paleognath birds and oviraptorid, dromeosaurid, and troodontid dinosaurs.

## CONCLUSIONS

Our study extends the origin of colored eggs from crown birds to oviraptorid dinosaurs. The result has important implications both for the origin of avian biology and the reproductive biology of theropods dinosaurs. This work also broadens the scope of paleontological research on molecular preservation and ecology to hard vertebrate tissues. Our study ties together previous hypotheses on the eumaniraptoran origin of partially open nesting, and paternal care. Also, potential future avenues for investigation are posed by the potential linkage between blue–green egg color and communal nesting, as well as polyandry, which represent yet unaddressed topics in extinct archosaurs.

The second aspect of our work focuses on its implications for molecular and soft tissue preservation through deep time. Chemically stable, relatively small biological molecules such as PP and BV appear to be protected from complete degradation over millions of years in carbonate biomineral matrices, in an oxidative sediment milieu. Similar biomolecule preservation may also be present in enamel, dentine and bone mineral. Ancient biomolecules and the soft tissues which they construct pave the way to trace life and its behaviors through time and, thus, invite further studies since they are easily detectable, more abundant than expected, and revolutionary in their ecological implications.

**Institutional abbreviations**

| | |
|---|---|
| **NMNS** | National Museum of Natural Sciences, Taichung, Taiwan |
| **PFMM** | Paleowonders Fossils and Mineral Museum, Taipei, Taiwan |
| **STIPB** | Steinmann Institute of Geology, Mineralogy, and Paleontology, Division of Paleontology, University of Bonn, Bonn, Germany |
| **ZFMK** | Zoologisches Forschungsinstitut und Museum Alexander Koenig, Bonn, Germany |

## ACKNOWLEDGEMENTS

We thank Y-N Cheng (NMNS), Y-F Shiao (PFMM), T Töpfer (ZFMK) and X Wu (CMN), who provided expertise and the eggshell specimens which are not from the collections of the Steinmann Institute. M Famulok (LIMES Institute, University of Bonn) provided additional laboratory facilities. We would like to acknowledge DEG Briggs (Yale University) for editing this manuscript, M Fabbri (Yale University) for proofreading the non-avian dinosaur statements made in this paper, and RO Prum (Yale University) for improving our ornithological inferences. Many thanks also to D Bartha (AMNH), G Mayer (Senckenberg Museum Frankfurt), DJ Varricchio (MSU), MA Norell (AMNH), J Vinther (University of Bristol), S Portugal (University of London), M Hauber (Hunter College), & all participants of the Society of Experimental Biology Eggshell Symposium 2015 for constructive discussions that improved this study. We would like to thank the Society of Vertebrate Paleontology and S Cohen for their interest and support of future research on the evolution of eggshell coloration and hard tissue taphonomy.

### Funding

The authors received no funding for this work.

### Competing Interests

The authors declare there are no competing interests.

### Author Contributions

- Jasmina Wiemann conceived and designed the experiments, performed the experiments, analyzed the data, wrote the paper, prepared figures and/or tables.
- Tzu-Ruei Yang conceived and designed the experiments, performed the experiments, analyzed the data, contributed reagents/materials/analysis tools, wrote the paper, prepared figures and/or tables.
- Philipp N. Sander conceived and designed the experiments.
- Marion Schneider performed the experiments.
- Marianne Engeser conceived and designed the experiments, performed the experiments, analyzed the data, contributed reagents/materials/analysis tools, reviewed drafts of the paper.
- Stephanie Kath-Schorr contributed reagents/materials/analysis tools.
- Christa E. Müller conceived and designed the experiments, analyzed the data, contributed reagents/materials/analysis tools, prepared figures and/or tables, reviewed drafts of the paper.
- P. Martin Sander conceived and designed the experiments, contributed reagents/materials/analysis tools, wrote the paper, reviewed drafts of the paper.

### Data Availability

The raw data are included in the preprint and the Supplemental File.

## Supplemental Information

Supplemental information for this article can be found online at http://dx.doi.org/10.7717/peerj.3706#supplemental-information.

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
