# Peer review of "Dinosaur origin of egg color: oviraptors laid blue-green eggs"

_PeerJ, doi:10.7717/peerj.3706_

## Round 0.1 · original submission · Major Revisions

I have received two thorough reviews and both reviewers feel the paper has strong potential. Reviewer 1 feels that parataxonomy using eggs is problematic since more conclusive statements can be made. I suggest some additional sentences in the intro or conclusions to address this point. Additionally, in particular reviewer 2 feels there are several technical details that need to be addressed. Additionally, this reviewer suggests (indeed requires) a photo of the eggs or eggshell fragments. In general reviewer 2 feels that, at this stage, the conclusions stretch too far beyond the data as presented here. I suspect with increased attention to detail the conclusions and supporting data will be better aligned with each other.

·

Basic reporting

See 'Comments to the authors'

Experimental design

See 'Comments to the authors'

Validity of the findings

See 'Comments to the authors'

Additional comments

Revision of the PeerJ manuscript -
Title: The blue-green eggs of dinosaurs: How fossil metabolites provide insights into the evolution of bird reproduction

I think this manuscript by Wiemann and colleagues is suitable for publication in PeerJ and will have high impact with important paleobiological implications. Although honesty I do not feel empowered to perform the English language due to I am not an Anglophone, but I can do some comments and improve as well some paleontological aspects of the manuscript.

General remarks:
Scientifically the manuscript is well structured, coherent and convincing with novelties palaeontological analyses. The manuscript is well organized and written in a clear and concise way. The manuscript title is perfect, reflecting clearly the work content, and the study, its findings and conclusions are clearly understood in the abstract.
I am not in favor of the egg parataxonomy, especially when this is used in relation to phylogenetic and/or evolutionary implications. However, I respect those who wish to do that. But I would like to clarify my point of view. Has been pointed by several authors (e.g., Arias et al., 1993, 2007; Nys et al., 2004; Grellet-Tinner, 2006, Grellet-Tinner et al., 2012; Hincke et al., 2012, among others), the eggshells –like the bone– are biomineralized systems, regulated by complex genetic-physiological processes and thus display phylogenetic information. The eggshell as a bioceramic structure deeply regulated by genes is an old vision and has been well studied. Hence, and despite current misleading trends (e.g., Mikhailov, 2013 or Fernández & Khosla, 2014), their inclusion in a parataxonomic classification, no matter for what purpose, is inappropriate and erroneous. A primary message on this regard supports previous observations that the egg parataxonomy is paraphyletic and the archosaurian oologic classification consists of unnatural groups (Grellet-tinner and Fiorelli, 2010; Grellet-Tinner et al., 2012a,b). Therefore, why use the parataxonomy classification and non-biological concepts when actually exists taxonomic certainty and phylogenetic resolution of this dinosaur eggs –like bones– with tremendous evolutionary implications? (in this case, oviraptorid Heyuannia eggs).
Although the manuscript should be published, I made some comments and revisions that must be addressed in the main text.

Specific comments
The specific comments are in the mains text (pdf file).

Final comment:
I like to point out that the manuscript is very interesting. Also, the manuscript is very precise, interesting, and well done, having a well development across the text. Have some interesting ideas and its well performed. The paper is sound, well argued, but some modifications are required. The illustrations and photography are well with good quality and with a logical sequence but should include one nice figure of the eggshell. The references are accurate but should include some of Steve Portugal and his group who work in this subject. So, and despite I am not a follower of the egg parataxonomy, the manuscript of Wiemann and col., is a very reasonable paper with valid interpretations, which is perfect and suitable for publication in PeerJ after consider the suggested revisions.

Reviewer 2 ·

Basic reporting

The paper would greatly benefit from better organization, with the data presented in Results and interpretations, based on that data, confined to Discussion and Conclusions rather than mixed throughout. For example, the first 6 lines under Results are their conclusions and these are stated before ANY supporting evidence has been given.

There appear to be no images of the actual eggshells studied.

There are inadequate citations as discussed below.

Experimental design

Pg 5. "Four pieces of eggshell were also taken from the blunt,middle,and acute parts of the egg "– this could be interpreted as 4 fragments were taken from each of the 3 regions, which I assume is not the case.

Not all methods for all analyses are included and some data from these analyses omitted from Results and not discussed. Sediment preparation, number of samples, etc. should be included in Methods. Same is true for extant eggshell. All results should be present, not just selective data – much is missing.

How were the fossil eggshells cleaned? These often have consolidant or other preservatives that could impact results, not to mention micro-organisms from the surrounding sediment.

The Methods do not include how pore length was measured. Is pore length considered as the same as the thickness of the shell when excluding ornamentation? Mention of ornamentation as a potential “pathway” makes this uncertain. Also, provide specimen numbers for the eggs in Methods – this is not clear except in the case of eggs from the Hongcheng basin.

Contamination

Protoporphyrins exist in microorganisms, particularly those that must synthesize heme for cytochrome production (aerobic microbes can use these). In fact, microbes can use iron containing porphyrin pigments to obtain metabolic iron, and microbes are plentiful in soils and sediments. More details on preparation of specimens would be helpful. Why aren’t graphs included for sediment sampled for biliverdin instead of only PP? According to the methods, both were analyzed (pg 8). There is also no explanation of how the sediments were prepared or how many were included.

I question the results of a 5 minute demineralization. In my experience, demineralization takes much longer. This seems too fast and extremely superficial, and thus values could perhaps come from superficial microbes. How do you rule out this possibility? The methods are also unclear about how the depths of the extracted PP and BV were determined and this would seem important to the conclusions drawn. Also, how far does the PP and BV extend into the calcite shell in modern eggs; most is concentrated in the cuticle and this is not preserved in your samples.

Validity of the findings

General Comments: The paper has potential for contributing to the literature but not in its present form. The conclusions extend far beyond the data, at least as it is presented here. Thin sections and SEM images of the eggshell understudy are missing. There are inadequate citations to the avian literature that would support interpretations (specific examples below). In addition, a key paper is omitted. Contrary to the manuscript, Igic et al. ( 2009) were the first to report PP and BV in fossil eggshell; they also report information that is in direct opposition to findings here, i.e., they report the ABSENCE of PP in emu eggshell. Further, although Kennedy and Vevers (1976) are cited in the manuscript, the fact that they report both PP and BV in many white eggs is omitted. The occurrence of both PP and BV in white eggs negatively impacts conclusions about strong blue-green color in dinosaur eggs. This needs to be discussed and explained. Finally, alternative hypotheses are not considered and this is very important given the common alteration of fossil eggshell over geologic time. More specific comments are given below.

Abstract

The abstract states the hypothesis (colored eggs in dinosaurs) before any supporting evidence is presented. This would be better placed after presentation of the data in the abstract.

Why is only PP ,and not BV, mentioned for supporting evidence of cuticle?

Introduction
p. 4 “egg colouration has attracted little study” – There have been extensive study of egg color. If the authors mean the chemistry of egg color, this should be reworded and provide references for these previous studies of avian eggs. This review should also include work on fossil eggshell color.

Define “immaculate” for the reader; this is not a common term for the paleo literature.

"While reptiles and monotreme mammals lack egg colouration because they bury their eggs.." – provide references, especially for monotremes burying their eggs.

Provide citation for open nests also being a synapomorphy for Neornithes or move the citation to the end of the sentence so that it includes both colored eggs and open nests, if that is the case.

More information about what is meant by positive and negative signaling would be helpful for readers who may not be familiar with the hypotheses.

Pg 5 Porosity measurements are used to calculated gas conductance, with the latter being more important to the paper; however, gas conductance is not mentioned here.

Identification of the eggs

In the abstract and elsewhere Cheng et al. assigned the embryonic eggs to M. yaotunensis or to “a closely related oviraptorosaurian”, yet this last is excluded from the current manuscript. An even bigger problem is that ALL of the eggs in this study are being assigned to M. yaotunensis without any comparisons to other oviraptorid eggs, differential diagnosis, or even a photograph of the eggshell structure that allows assessment of this conclusion. This taxonomic assignment to M. yaotunensis is then used to support the presence of a cuticle, even though cuticle does not occur in the specimens under study. These multiple tiers of inference requires a lot of faith on the part of the reader!

Geology

"The geological background is discussed in detail below." -- Providing the names of 3 formations from which the eggs were recovered in Fig 1 is NOT a detailed discussion of geologic background. Geologic background would be better included in a separate geology section, rather than scattered about in the text and figure caption.

Results

There appear to be 2 data sets (PP and BV) for the emu eggshell. If so, please include results of both. Why aren’t the methods used for these extant shells included under “Pigment extraction and detection”, rather than placed in Results? Were they handled the same way?

Similarly, if sediment samples were analyzed for each of the 3 eggs, why are the results limited to one? Please include all and explain in the Methods how the sediment samples were processed and give sample numbers.

The first 6 lines of Results are conclusions and discussion and these are presented before ANY supporting data. Why is the extant material discussed here but not the fossil?


Pg 7 This is not the first detection of PP and BV in fossils (see Igic et al. 2009). More importantly, Igic et al also reports the absence of protoporphyrin in emu egg. This finding should be discussed because it brings your methodology into question because you are comparing the fossil to the extant eggshell. In addition, their methods seem to provide greater support since they used 2 different analyses, rather than one. On page 10, you include values of PP and BV for Sturnus vulgarus eggs; are there none available from the literature for emu, particularly if PP exists in these eggs?

Most importantly, Kennedy and Vevers (1976) report the presence of both PP and BV in many white eggs. How, then, can green-blue color be established in the dinosaur eggs and white color be ruled out? A ratio between the PP and BV is mentioned in your manuscript as evidence that the dinosaur eggs are “strongly colored”. However, there are no citations of the avian literature that would validate this ratio for color determination. How much PP and BV does it take to produce green-blue color? How do you account for white eggs with both PP and BV? Further, there is also no citation for the “greater chemical stability” of PP than BV. How do you know this? Please provide references.

Diagenesis

There is no discussion of the potential influence of diagenesis on preservation of PP and BV. Igic et al (2009) used two different methods and detected PP in 2 samples of Moa eggs and none in another Moa egg from a different locality, concluding that " .. patterns of pigment detection are due to environmental contaminants or chemical and physical changes to the pigments on and inside the eggshell matrix that occurred during the unknown period between egg formation and laying and the time of pigment extraction in this study.”
.
Again, how has diagenetic alteration been ruled out? Providing a hypothetical color for these eggs in Fig 1 also seems highly questionable. CL images would be helpful in assessing diagenesis. In fact, no images of the actual eggshell used for GH2O, ootaxonomic assignment , and chemical analyses are included in the manuscript or supplemental material. Examples of a radial and tangential section are essential.

Cuticle - Contrary to the manuscript, cuticle has been reported previously in Mesozoic eggs (Schweitzer et al. 2002; Jackson and Varricchio 2010, to name two).

The image in the supplemental material is inadequate to demonstrate the presence of a cuticle, even if these are the same oospecies. An enlargement is needed, especially since this paper is not yet published. The dashed line appears to represent the contact between two generations of secondary calcite growth. The purported “cuticle” shows little or no similarity to that in modern birds in either thickness or morphology. This difference, compared to modern cuticle, should be discussed.

Pg 10 - "These concentrations represent visible amounts of PP and BV " – if “visible” refers to the ability to detect color of the egg, this needs a supporting citation from the avian literature about what concentrations are necessary. If it means measurable amounts of PP and BV, the wording should be changed.

Discussion

Alternative hypotheses and diagenesis should be discussed.

Pg 12 “ …. earliest evidence of open nesting is found in the most bird-like theropods, the maniraptoran dinosaurs”. This is incorrect. These nests are partially open nests and differ significantly from open nests of birds. I am not aware of any bird today that lays partially buried eggs in sediment that cannot be turned.

"..lends support to the hypothesis of paternal care in dinosaurs (Varricchio et al., 2008)." Parental care in dinosaurs was proposed by Horner long before Varricchio et al. and this citation should be included. Alternatively, change to “theropods”.

"….dispensable for the maternal metabolism" - this doesn’t make sense. Do you mean that BV production results from maternal metabolism?

"Pigment incorporation, especially of BV, into the crystalline matrix implies a layered formation of the eggshell (in manaraptorans)" -- how does the presence of PP and BV in the cuticle and upper part of the eggshell tell you anything about the crystalline structure below it?

Figures

Far too much discussion is included in the figure captions rather than in the text; as a result, this information often does not include proper citations. Use the figure captions to explain what the reader is looking at in the figures and discuss it in the Discussion section.

Fig. 1
(D) This is called a “partially open nest” in C, which is appropriate since the bottom layer would have little or no contact with an adult. This terminology should be consistent throughout the manuscript, rather than alternating back and forth between “partially open” and “open nest”. Many of the fossil bird eggs cited are also partially buried, occur singly (rather than in clutches), and have no evidence of a nest structure. To say they are “partially open nests” is misleading.

Fig. 3. All archosaurs (including crocodilians) have fully calcified eggs so this attribute did not “evolve in Dinosauria”. Fully calcified eggs also occur in some turtles and geckos as well.

Contrary to the figure caption, there is little similarity between clutch structure in oviratorids and troodontids; the latter are more upright in the nest, in a far more compact arrangement. As mentioned above, the speculation about egg color in Troodon does not belong in a figure caption and should be moved to the Discussion and, if included at all, the type of nest structure for this taxon should be properly referenced.

Fig 3 (C) For the benefit of those who are unfamiliar with fossil eggs, it should be stated in the caption that the color of the fossil eggs in C results from diagenesis, not original pigment color.

Conclusions

Like the figure captions mentioned above, this section includes information that is not presented in the paper and/or lacks citations. For example, the “chemical stability” of PP and BV (not referenced) and “oxidative chemical milieu in the embedding sediment”, which has not been discussed previously nor is data presented in Results that supports this statement.

---

## Round 0.2 · Minor Revisions

I sent your paper out for review again because it had been a long time since the first reviews came and because the revisions were fairly extensive. The reviewer only has few minor suggestions to improve the paper.

·

Basic reporting

no comment

Experimental design

no comment

Validity of the findings

no comment

Additional comments

The manuscript entitled Dinosaur origin of egg color: oviraptors laid bluegreen eggs by Wiemann and colleagues provides strongly support evidence for the presence of egg coloration in non-avian dinosaur eggs, and represents an important contribution not only for palaeoology, but for palaeontology in general, as states the preservation potential of certain biological molecules that can survive fossilization.
I read the original preprint manuscript when it came out more than a year ago, and I am pleased to see how much the already good manuscript has improved through revision. I think the authors present strong evidence for the presence of coloration in oviraptorid eggs, and thoroughly discuss all possible causes and implications of this trait for the breeding and nesting behaviour of this group of non-avian dinosaurs. I strongly recommend the publication of this work, although I recommend a few additional revisions -presented in order of importance-, that do not compromise the main conclusions of this excellent piece of research.
Lines 432-434. This is not a conclusion of your research, as you do not provide evidence of egg arrangement in nest, only present water vapor conductance calculations that point towards partially open nest. Furthermore, this has been previously stated in the literature (see Huh et al 2014, Pu et al 2017 and references within). This paragraph should be deleted.
Supplementary information section F
Here you present a description of the cuticle inner layer based only in the observation of a single radial thin section under a petrographic microscope. With the data presented in your manuscript, the additional layer can be interpreted in several different ways, from a cuticle to an overlaying diagenetic cement. The only evidence of a putative cuticle is the high concentration of PP, but this can be also explained by replacement of the original cuticle, of mobilization of PP from the eggshell to the diagenetic layer during fossilization. A mineralogical, chemical and taphonomical analysis (including SEM and cathodoluminescence, and maybe epifluorescence) is needed to assess the presence of the cuticle. I would suggest removing this section if this analysis cannot be carried at his moment.
Lines 360-361. I do not understand why taphonomy must affect concentrations of both molecules equally, as they have completely different solubility and chemical properties. What I understand is that taphonomy may have affect both molecules, but not necessary in the same factor or direction (e.g. increasing the concentration of one of them while dismissing the other). Please reword this sentence.
Lines 129-135 should be moved to the methods section, to cut an otherwise long introductory section.
Finally, I want to support the authors in their use of parataxonomy. Even if some of the eggs included in this manuscript can be related to a taxon, it is highly probable that relatively close oviraptorids laid similar eggs, that wold be classified as the same ootaxon. Parataxonomy is an elegant solution to this problem, and some recent studies seem to forget this point.
I think this paper opens a new research area in palaeoology. There is much more to explore after this initial description of the technique. I would have love to see a complementary analysis of both actual and fossil uncoloured eggshells, that would also strength the results of this research. I encourage the authors to continue this line of investigation and further explore this topic.

Miguel Moreno Azanza

---

## Round 0.3 · accepted · Accept

Thank you for substantively addressing the reviewer's comments.